# LimoRhyde2: Genomic analysis of biological rhythms based on effect sizes

**Dora Obodo**[ID][1,2�)], **Elliot H. Outland**[1☉], **Jacob J. Hughey**[ID][1,2,3]*

**1** Department of Biomedical Informatics, Vanderbilt University Medical Center, Nashville, Tennessee, United States of America, **2** Program in Chemical and Physical Biology, Vanderbilt University School of Medicine, Nashville, Tennessee, United States of America, **3** Department of Biological Sciences, Vanderbilt University, Nashville, Tennessee, United States of America

☉ These authors contributed equally to this work.
* jakejhughey@gmail.com

## Abstract

Genome-scale data have revealed daily rhythms in various species and tissues. However, current methods to assess rhythmicity largely restrict their focus to quantifying statistical significance, which may not reflect biological relevance. To address this limitation, we developed a method called LimoRhyde2 (the successor to our method LimoRhyde), which focuses instead on rhythm-related effect sizes and their uncertainty. For each genomic feature, LimoRhyde2 fits a curve using a series of linear models based on periodic splines, moderates the fits using an Empirical Bayes approach called multivariate adaptive shrinkage (Mash), then uses the moderated fits to calculate rhythm statistics such as peak-to-trough amplitude. The periodic splines capture non-sinusoidal rhythmicity, while Mash uses patterns in the data to account for different fits having different levels of noise. To demonstrate LimoRhyde2's utility, we applied it to multiple circadian transcriptome datasets. Overall, LimoRhyde2 prioritized genes having high-amplitude rhythms in expression, whereas a prior method (BooteJTK) prioritized "statistically significant" genes whose amplitudes could be relatively small. Thus, quantifying effect sizes using approaches such as LimoRhyde2 has the potential to transform interpretation of genomic data related to biological rhythms.

## Introduction

Much of life on Earth shows rhythms that follow the ~24-hour cycle of day and night. To produce these daily rhythms, each organism has a system of cell-autonomous oscillators, or circadian clocks, that senses environmental cues and drives cellular, physiological, and behavioral outputs [1]. In mammals, these clocks "tick" in nearly every tissue [2], although their tissue-specific mechanisms and inter-tissue interactions are only partially understood [3–6]. To study circadian systems and their relevance to human health, genome-scale approaches are invaluable, e.g., revealing mechanisms of intercellular communication in the circadian response to feeding [7] and highlighting drug targets for circadian medicine [8].

Nonetheless, current methods for analyzing rhythms in genome-scale data have multiple limitations. Some methods require that timepoints be equally spaced, or assume that rhythms are sinusoidal. Most importantly, almost all current methods—including our method

**Data Availability Statement:** The LimoRhyde2 R package is available at https://limorhyde2.hugheylab.org. Code, data, and results for this study are available on Figshare (https://doi.org/10.6084/m9.figshare.22001519).

**Funding:** HHS | NIH | National Institute of General Medical Sciences (NIGMS):Jacob J Hughey R35GM124685; HHS | NIH | National Institute of General Medical Sciences (NIGMS):Dora Obodo F31GM143909. The funders had no role in study design, data collection and analysis, decision to publish, or preparation of the manuscript.

**Competing interests:** No, there is no conflict of interest. My manuscript contains the following statement: "The authors declare that they have no conflict of interest".

LimoRhyde [9]—focus on hypothesis testing (i.e., p-values and statistical significance), raising at least two issues [10, 11]. First, the null hypothesis, e.g., that a gene has a log fold-change of exactly zero, is seldom true. Rather, observed biological effects typically vary from small to large. Second, statistical significance does not necessarily imply that a result is biologically relevant, since the p-value depends on both the estimated effect size and its uncertainty.

An alternative to calculating p-values is to estimate effect sizes for rhythmic properties (e.g., amplitude and phase) directly. The broader field of genomics has developed multiple methods for estimating effects (e.g., log fold-change) [12–14], which has become an important part of differential expression analysis. Previous work has applied effect size estimation to biological rhythms [15], but not to genome-scale data, while other work has incorporated confidence intervals, but only using cosinor regression on a small set of clock genes [16].

Therefore, we developed LimoRhyde2, a new approach to quantify rhythmicity in genomic data. LimoRhyde2 integrates and builds on state-of-the-art tools and practices to rigorously analyze data from genomic experiments [9, 17, 18], capture non-sinusoidal rhythms [19], and accurately estimate effect sizes [14, 20]. Whereas prior methods to analyze rhythmic data seek to answer the question "Is there an effect?", LimoRhyde2 seeks to answer the often more relevant question "How strong is the effect?". To illustrate LimoRhyde2's utility, we applied the method to multiple circadian transcriptome datasets, comparing its output to that of a prior method. Our findings suggest that LimoRhyde2 can enable new insights into biological rhythms and circadian systems.

## Methods

The LimoRhyde2 R package is available at https://limorhyde2.hugheylab.org. Code, data, and results for this study are available on Figshare (https://doi.org/10.6084/m9.figshare.22001519).

### LimoRhyde2 algorithm

**Fit linear models.** Similarly to LimoRhyde, LimoRhyde2 starts by fitting a linear model to the measurements of each genomic feature in the dataset (e.g., expression of each gene). By default in LimoRhyde2, the model terms for time (e.g., zeitgeber or circadian time) are based on a periodic cubic spline with three internal knots. Alternatively, the terms for time can be based on sine and cosine components (i.e., cosinor, the default in LimoRhyde). The model can include additional terms for covariates, e.g., to account for batch effects, but assumes all samples come from the same condition. Thus, the model could be

$$E\left(y_{g,i}\right) = \beta_{g,0} + \sum_{j=1}^{n} \beta_{g,j} B_j(\theta_i)$$

Where $E(y_{g,i})$ is the expected (log-transformed) measurement for feature $g$ in sample $i$, $\beta_{g,j}$ are the unknown coefficients for feature $g$, $n$ is the number of spline knots, $B_j$ are the periodic spline basis functions with period $\tau$, $\theta_i = \frac{2\pi t_i}{\tau}$, and $t_i$ is the time for sample $i$. LimoRhyde2 fits the models, i.e., estimates the coefficients, using limma-trend, limma-voom [25, 26], or DESeq2 [12], all state-of-the-art approaches for analyzing genomic data.

In initial testing we observed that the fitted curves of the periodic spline sometimes varied noticeably depending on the locations of the spline knots, particularly for more rhythmic genes. To avoid this behavior and make the fits more robust if the linear model is based on periodic splines (not cosinor), LimoRhyde2 repeats the above procedure multiple times, fitting a series of models for each feature such that the locations of the knots in each model are shifted by a different amount. For $m$ shifted models (default 3), the value of the shift $d_k$ for model $k$ is

set to

$$d_k = \frac{(k-1) \cdot \tau}{m \cdot (n+1)}$$

The overall raw fit $f_{g,k}(t, \ldots)$ for feature $g$ is calculated as

$$f_g(t, \ldots) = \frac{1}{m} \sum_{k=1}^{m} f_{g,k}(t, \ldots)$$

where $f_{g,k}(t, \ldots)$ indicates the expected measurement of feature $g$ according to model $k$, as a function of time $t$ and any covariates.

**Moderate model coefficients.** LimoRhyde2 then moderates the model coefficients to obtain posterior fits using multivariate adaptive shrinkage (Mash) [14], which uses Empirical Bayes to learn patterns of similarity between coefficients and to improve estimates of effect sizes. LimoRhyde2 runs Mash on the coefficients for time for all shifted models. LimoRhyde2 does not moderate the intercept coefficients, as the relatively large number of samples in typical circadian transcriptome experiments makes these coefficients' standard errors (and the effect of Mash) quite small. By default, LimoRhyde2 runs Mash with data-driven covariance matrices, computed based on principal component analysis of strong signals in the data, with the number of principal components set to the number of spline knots. Given the raw estimates and standard errors for each coefficient for each feature, Mash computes corresponding posterior distributions, including posterior means and standard deviations. Mash's approach to estimating posteriors takes the place of the usual multiple testing adjustment, e.g., estimation of false discovery rates.

**Calculate rhythm statistics.** LimoRhyde2 then uses the moderated coefficients (or optionally, the raw coefficients) to calculate the following rhythm statistics, i.e., properties of the fitted curve with respect to time between 0 and $\tau$ of each feature:

- mesor (mean value)

- peak or maximum value

- peak phase (time at which the peak value occurs)

- trough or minimum value

- trough phase (time at which the trough value occurs)

- peak-to-trough amplitude (peak value minus trough value)

- root mean square (RMS) amplitude, calculated as $\sqrt{\frac{1}{\tau} \int_0^\tau (f(t) - E[f(t)])^2 \, dt}$, where $E[f(t)] = \frac{1}{\tau} \int_0^\tau f(t) dt$

**Quantify uncertainty.** To quantify uncertainty in the fits, LimoRhyde2 can draw samples from the posterior distributions computed by Mash. For each posterior sample, which corresponds to a set of possible model coefficients for each feature, LimoRhyde2 can calculate the expected measurement of each feature as a function of time (and possibly any covariates), as well as the corresponding rhythm statistics. LimoRhyde2 then uses the resulting posterior distributions to calculate quantities such as the 90% (equal-tailed or highest-density) credible interval, the Bayesian analogue of a confidence interval.

Because amplitude estimates are non-negative, a credible interval based only on the posterior samples' amplitudes would nearly always be strictly positive (implying that that feature is rhythmic). This would preclude the credible interval from crossing zero, even if the rhythm were quite weak and the posterior samples' phases were highly variable. Thus, LimoRhyde2 constructs credible intervals for peak-to-trough amplitude and RMS amplitude by first changing the sign (from positive to negative) of amplitudes for posterior samples whose peak phase is greater than $\frac{\tau}{4}$ away in either direction from the circular mean peak phase (weighted by amplitude). In this way, the credible interval for a rhythm whose amplitude is highly uncertain can span zero.

## Processing circadian transcriptome data from mice

For microarray data from mouse lung (GSE59396), we used the seeker R package [27] to download the sample metadata and processed (Illumina) expression data from NCBI GEO, map probes to Entrez Gene IDs [28], and return $\log_2$-transformed expression values. For RNA-seq data from mouse liver (GSE67305) and suprachiasmatic nucleus (SCN) (GSE72095), we used seeker to download the sample metadata and to download and process the raw reads. We processed the data using Trim Galore for adapter and quality trimming [29], FastQC [30] and MultiQC [31] for quality control, and salmon [32] and tximport [33] for quantifying gene-level counts and abundances based on Ensembl Gene IDs. We obtained the transcriptome index for salmon using refgenie [34]. Based on the plotMDS function from the limma package, we removed from analysis one extreme outlier sample from GSE59396. For GSE67305 and GSE72095, we kept only those genes having counts per million (CPM) $\geq 0.5$ in at least 75% of samples (irrespective of timepoint). To avoid unrealistically low log-transformed CPM values and artificially inflated effect size estimates, for each sample-gene combination that had zero counts, we impute the counts as the minimum of the non-zero counts across all samples for that gene.

## Quantifying rhythmicity using LimoRhyde2

We ran LimoRhyde2 using three knots (or using a cosinor model where noted), a period of 24 h, and either limma-trend (for GSE59396) or limma-voom (for GSE67305 and GSE72095). Where applicable, we calculated 90% equal-tailed credible intervals based on 200 posterior samples.

## Detecting rhythmicity using BooteJTK

To compare performance of a prior method for detecting rhythmicity in genomic data, we used BooteJTK (source code from https://github.com/alanlhutchison/BooteJTK). BooteJTK performs hypothesis testing for rank-order correlation between a feature's time series and a set of reference waveforms, using parametric bootstrapping to account for measurement uncertainty. We ran BooteJTK using the default settings. In particular, we generated 25 bootstrap resamples of each gene's time series, used a period of 24 h, treated samples collected 24 h apart as replicates, and searched for phases and asymmetries at 2-h intervals from 0 to 22 h. For each dataset, we passed BooteJTK the log-transformed microarray expression values or the $\log_2(\text{cpm} + 1)$ values for the same set of genes that we passed to LimoRhyde2. We adjusted the resulting p-values using the Benjamini-Hochberg (BH) method to control the false discovery rate [35]. To quantify agreement between LimoRhyde2 and BooteJTK in ranking rhythmic genes, we calculated Cohen's kappa using the irr R package. To quantify the circular correlation of the two methods' phase estimates in each dataset, we used the circ.cor2 function of the

Directional R package, and included only those genes having a BooteJTK adjusted p-value $\leq 0.2$ and a LimoRhyde2 amplitude in the upper 10 percent.

## Results

To demonstrate how LimoRhyde2 quantifies rhythmicity, we applied it to transcriptome datasets from a range of tissues, experimental designs, and measurement techniques [36–38] (Table 1). We used LimoRhyde2 to fit periodic spline-based linear models for each gene (raw fits), moderate the raw fits using Mash (producing posterior fits), and calculate each gene's rhythm statistics based on the raw and posterior fits (Fig 1). As expected, the posterior fits tended to have lower rhythm amplitudes compared to the raw fits (Fig 2), and higher standard errors in the raw fits led to greater amplitude reduction in the posterior fits for many genes (S1 Fig). In addition, the periodic splines allowed LimoRhyde2 to fit rhythms that appeared non-sinusoidal, avoiding the occasional poor fit of the cosinor model (S2 Fig).

To compare LimoRhyde2 to a prior method for detecting rhythmicity, we analyzed the same data using BooteJTK [21], a refinement of the popular JTK_CYCLE [39] that accounts for measurement uncertainty. Like other prior methods, BooteJTK calculates a p-value (an estimate of the probability that a given feature is *not* rhythmic) for each genomic feature, which can then be used for ranking. As LimoRhyde2 does not calculate p-values, but instead estimates a gene's rhythm, a convenient way to use its output to rank genomic features is by posterior amplitude. Importantly, the raw fits do not account for standard error and are therefore unreliable indicators of true effect size.

Across the three datasets, we found the adjusted p-values from BooteJTK were only modestly correlated with the amplitudes from LimoRhyde2 (Fig 3A; mean Spearman correlation 0.72). Whereas BooteJTK prioritized monotonic rhythms with high signal-to-noise but perhaps relatively low amplitude, LimoRhyde2 prioritized high-amplitude rhythms of various shapes (Fig 3B). Accordingly, the two methods showed relatively weak agreement in the top-ranked genes in each tissue (Fig 3C). In contrast, the two methods' phase estimates were highly correlated (mean circular correlation 0.86). The two methods differed markedly in runtime: 2 minutes for LimoRhyde2 and 73 minutes for BooteJTK (mean per study, LimoRhyde2 run in parallel on 6 cores, BooteJTK does not run in parallel).

As a typical use case, we further analyzed the posterior fits and statistics from LimoRhyde2 for each dataset. Here, we also used LimoRhyde2's ability to sample from the posterior distributions to quantify uncertainty as 90% credible intervals (Fig 4). As expected, a noisy fit, such as that of Nr1d1 in the SCN, led to a credible interval for amplitude that spanned zero (Fig 4A–4C). Consistent with prior work [40], most clock genes showed high amplitudes in each tissue (S3 Fig), although some of the highest-amplitude genes were tissue-specific (Fig 4D). Contrary to previous findings [41], the highest-amplitude genes tended to be moderately expressed, not the most highly expressed (S4A Fig). Furthermore, among the top 25% of genes by amplitude, the joint distribution of amplitude and phase, as well as the marginal distribution of phase, differed widely by tissue (S4 Fig). Overall, these results illustrate the value of LimoRhyde2's approach to quantifying rhythmicity.

**Table 1. Characteristics of the three circadian transcriptome datasets used for validation.**

| Study | Reference | Platform | Tissue | Interval (h) | Num. of biological samples | Light-dark regimen |
|---|---|---|---|---|---|---|
| GSE59396 | [36] | microarray (Illumina beadchip) | lung | 4 | 36 | LD 12:12 |
| GSE67305 | [38] | RNA-seq (100 bp single-end) | liver | 2 | 24 | LD 12:12 |
| GSE72095 | [37] | RNA-seq (100 bp paired-end) | suprachiasmatic nucleus (SCN) | 4 | 18 | LD 12:12 |

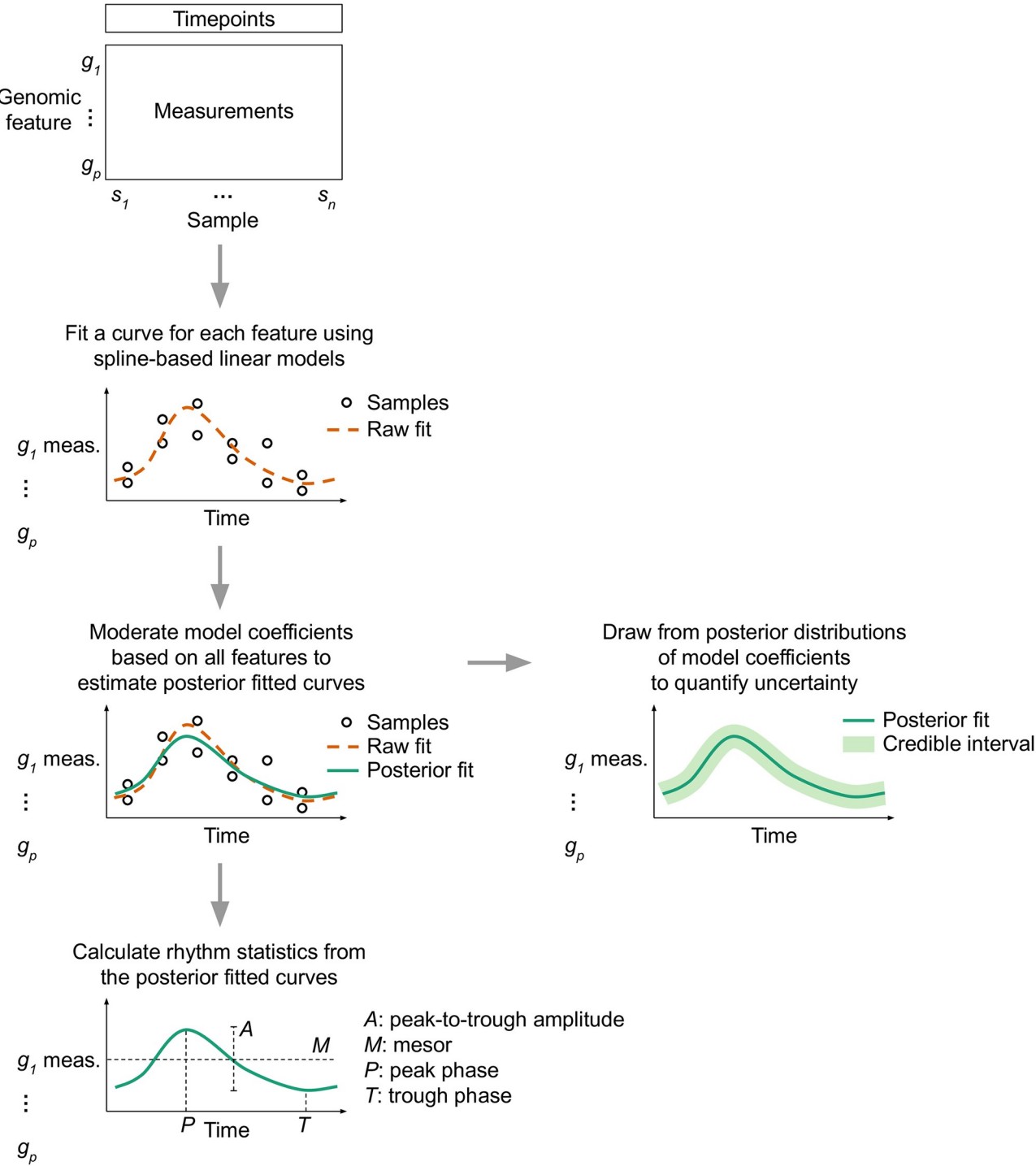

**Fig 1. Schematic overview of LimoRhyde2's approach to quantifying rhythmicity in genomic data.** Given a genomic feature (row) by sample (column) matrix of measurements, LimoRhyde2 fits a curve (*dashed orange line*) based on periodic splines describing how each feature's measurements change as a function of time. To adjust for noise and uncertainty in the fits, LimoRhyde2 uses Mash to moderate the model coefficients, yielding a posterior fit (*solid green line*) for each feature. Using the posterior fits and their distributions, LimoRhyde2 calculates rhythm statistics and credible intervals.

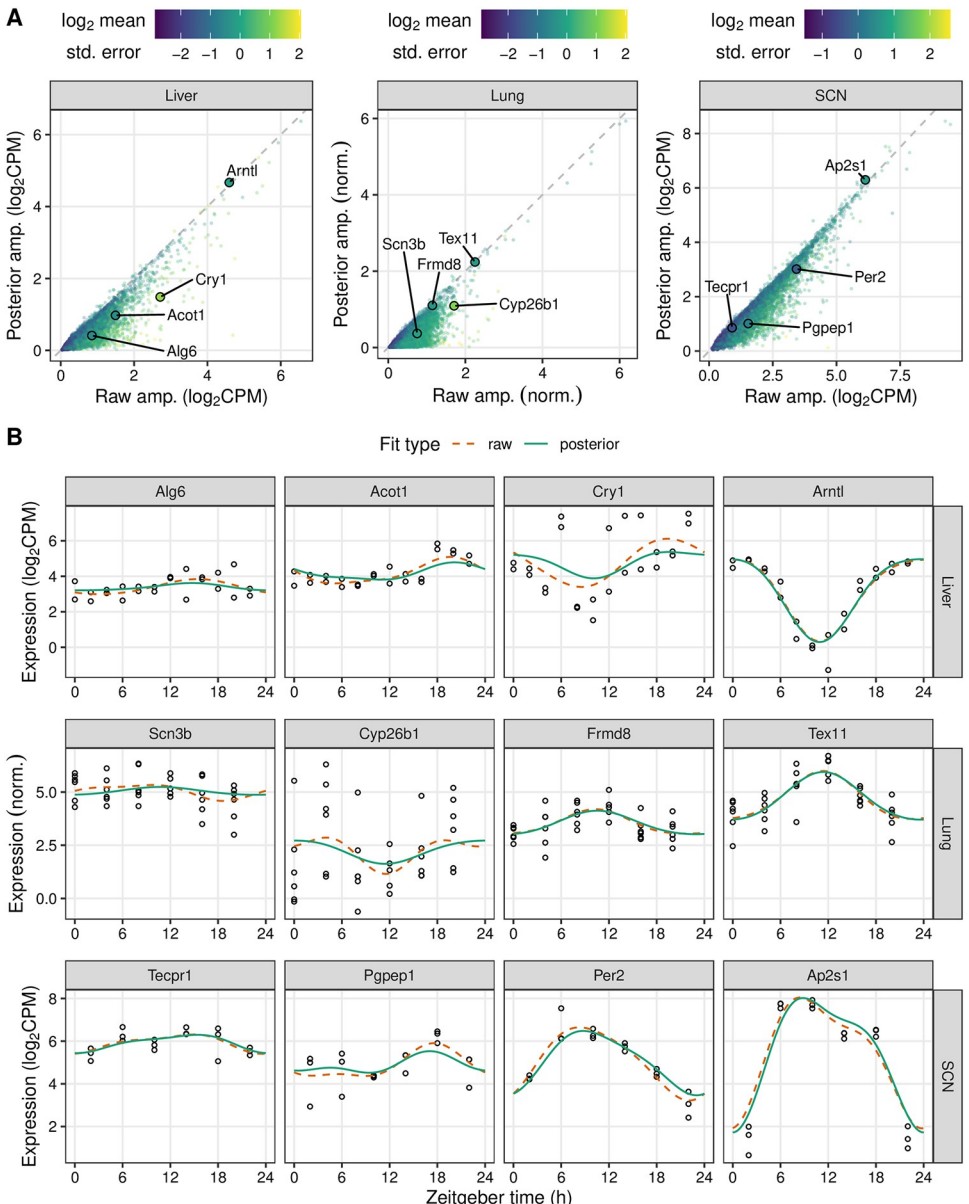

**Fig 2. LimoRhyde2 uses raw fits and their standard errors to obtain posterior fits. (A)** Scatterplots of posterior peak-to-trough amplitude vs. raw peak-to-trough amplitude for transcriptome data from mouse liver (GSE67305), lung (GSE59396), and SCN (GSE72095). *Points* represent genes, *color* represents log$_2$ mean standard error of the gene's raw fit. *Dashed lines* indicate y = x. **(B)** Time-courses of expression of genes (sorted by posterior amplitude) labeled in (A) in the respective tissue. *Curves* represent fits calculated by LimoRhyde2. *Points* represent samples.

## Discussion

Genomic data have yielded valuable insights into circadian systems and their relevance to human health, but previous methods for genomic analysis of biological rhythms have multiple limitations, perhaps the most severe being an overreliance on p-values and statistical significance.

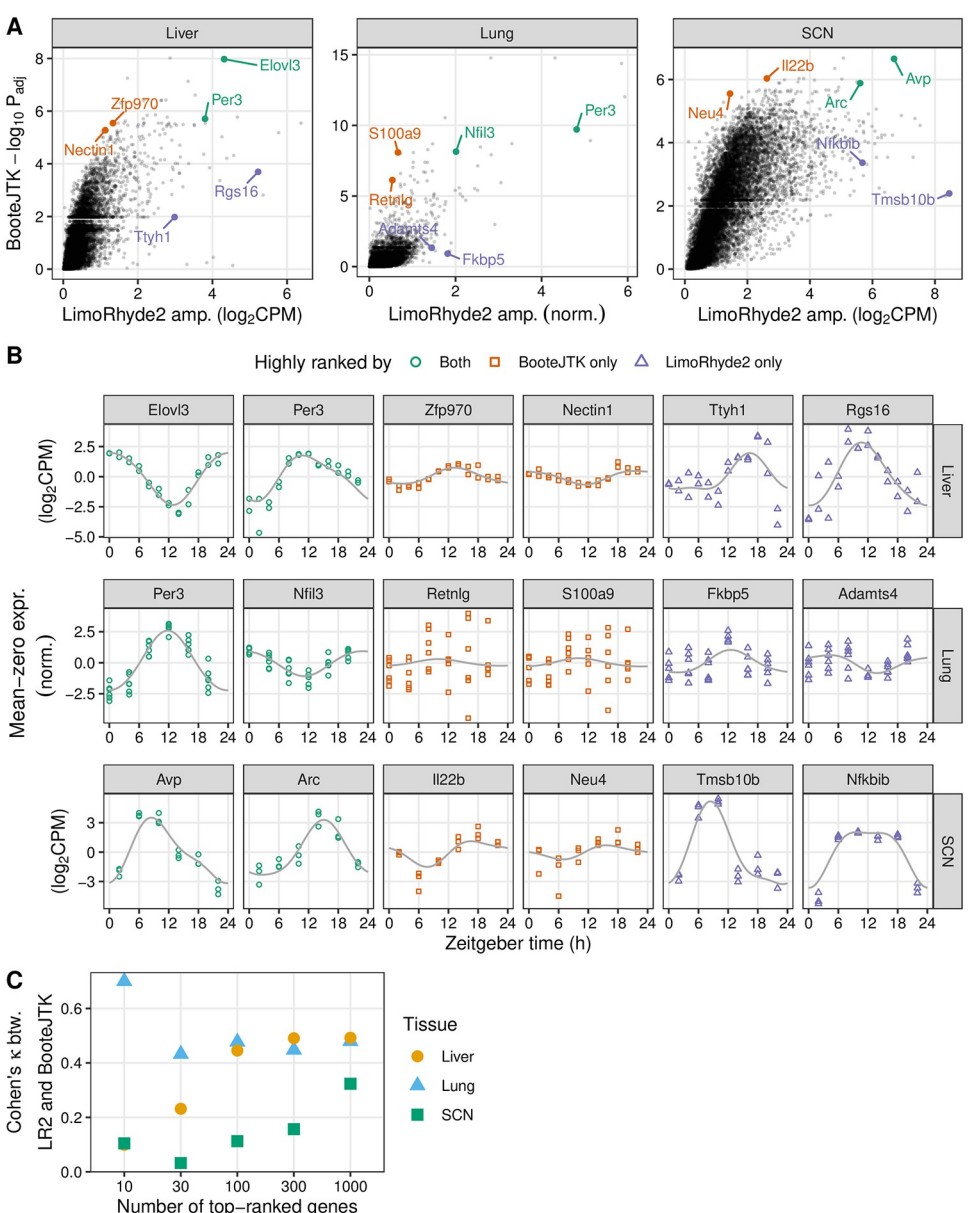

**Fig 3. LimoRhyde2 prioritizes high-amplitude rhythms, compared to BooteJTK. (A)** Scatterplots of -log₁₀ adjusted p-value calculated by BooteJTK vs. posterior peak-to-trough amplitude calculated by LimoRhyde2 for each tissue (indicated at *top*). *Points* represent genes. Genes towards the top are prioritized by BooteJTK, genes to the right are prioritized by LimoRhyde2. **(B)** Time-courses of expression of genes (indicated at *top*) labeled in (A) in the respective tissue (indicated at *right*). *Points* represent samples. *Colors* represent relative ranking of the genes according to the two methods. *Curves* represent posterior fits calculated by LimoRhyde2. Each gene's expression is centered at zero to highlight differences in amplitude. **(C)** Interrater agreement, as quantified by Cohen's κ, of top-ranked genes based on LimoRhyde2 posterior amplitude and BooteJTK -log₁₀ adjusted p-value. *Shape* and *color* represent tissue.

LimoRhyde2's posterior estimates—fitted curves, resulting statistics, and credible intervals —account for uncertainty in the raw estimates and for different patterns of rhythmicity. This aspect of LimoRhyde2 is enabled by Mash [14], which uses shrinkage to share information among genomic features. This shrinkage of the coefficients and their standard errors is distinct

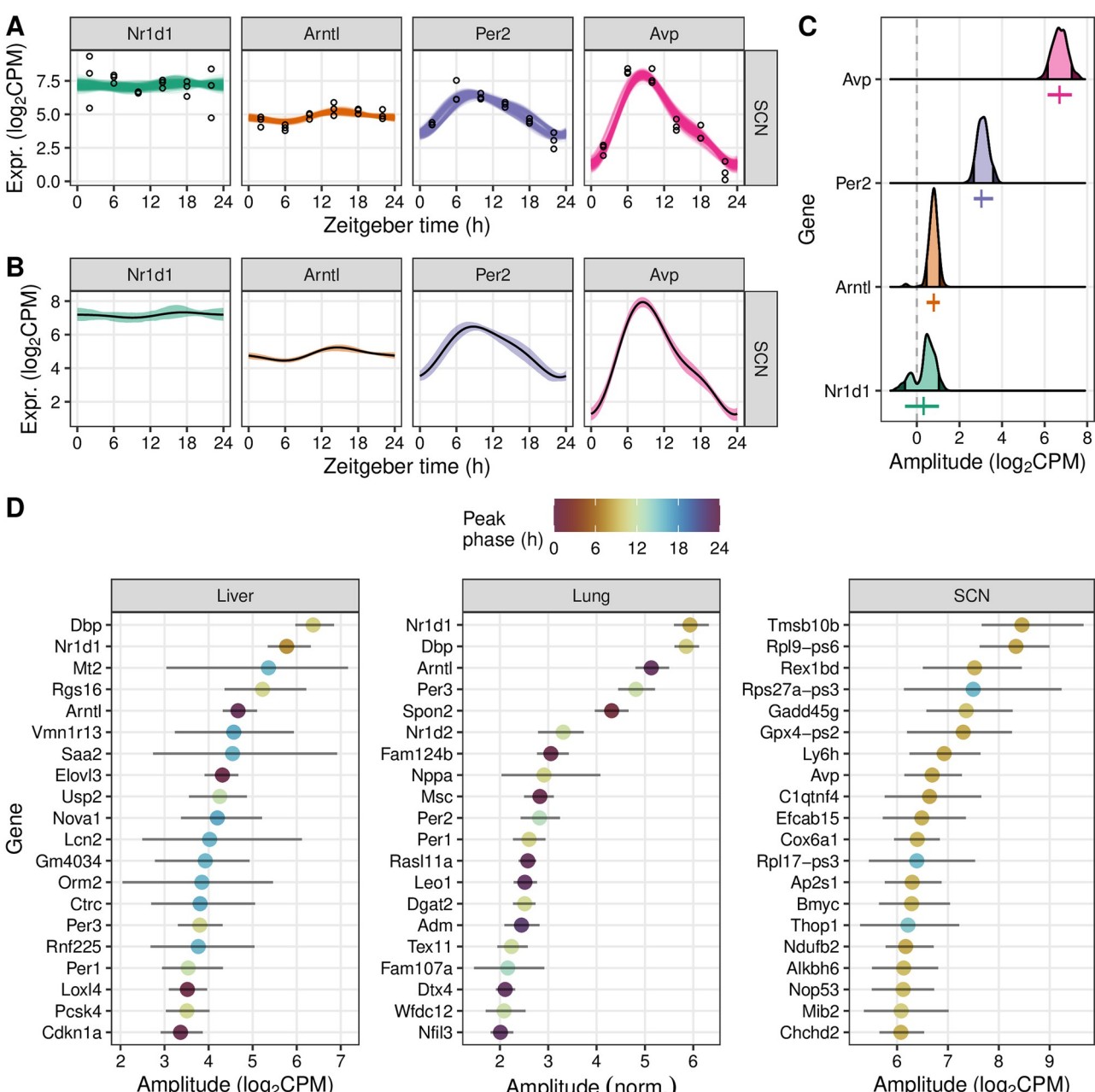

**Fig 4. LimoRhyde2 quantifies uncertainty in rhythmicity using credible intervals. (A)** Time-courses of expression for selected genes (indicated by *color* and at *top*) in the SCN. *Points* represent samples. *Curves* represent fits from 200 draws from the posterior distributions. **(B)** Time-courses of expression, where *lines* represent posterior means and *ribbons* represent 90% credible intervals calculated from the posterior draws. **(C)** Density plots of peak-to-trough amplitude based on the posterior draws. *Dashed line* indicates 0 amplitude. *Shaded regions* represent lower and upper bounds of the 90% credible intervals. *Vertical colored lines* represent posterior means of amplitude. *Horizontal colored lines* represent 90% credible intervals. **(D)** Amplitudes and corresponding 90% credible intervals for the top 20 genes ranked by amplitude in each tissue (indicated at *top*). *Color* represents peak phase.

from that applied by limma to residual variances, which in practice have little effect given the relatively large sample sizes of circadian experiments. Such sharing by Mash may reduce goodness of fit for any one genomic feature, but is a well-validated strategy to prevent overfitting and improve estimates [14, 20], while avoiding the danger that "statistically significant" results

can imply overestimated effect sizes [42]. Furthermore, LimoRhyde2's approach bypasses the need to consider both raw amplitude and p-value, which would require another arbitrary cut-off [43, 44]. Thus, in LimoRhyde2, genome-scale data are neither a burden (for multiple testing) nor a curse (of dimensionality), but rather, an advantage.

Traditionally, efforts to understand the core circadian clock in various species have focused less on rhythm amplitude than on period and phase. However, the goal of genomic studies is often not to understand the core clock itself, but rather to determine how the clock and daily rhythms influence physiology. In this respect, amplitude may be as relevant as period and phase. For example, a protein whose mRNA shows a rhythm amplitude of 3 $\log_2$CPM may be a more promising target for circadian medicine than one whose mRNA shows an amplitude of only 0.3.

Given the distinct goals of LimoRhyde2 compared to previous methods (Table 2), the relevant differences between the methods' output are not quantifiable in terms related to binary classification (precision, recall, false positive, etc.). Therefore, we opted to analyze real circadian transcriptome data rather than simulated data, whose generation requires many simplifying assumptions. Although the true effect sizes in these data are unknown, our results indicate that LimoRhyde2 efficiently prioritizes large effects that have functional significance in the circadian system yet would have been underappreciated by methods dependent on p-values and statistical significance.

For example, among the genes ranked considerably higher by LimoRyde2 (amplitude) than by BooteJTK in the liver were Rgs16, Dhrs9, and Mt2. Rgs16 regulates daily rhythms of G-protein signaling in the SCN [45] and substrate oxidation in hepatocytes [46]. Dhrs9 belongs to a set of genes whose expression forms a robust biomarker of internal circadian time from human blood [47]. Mt2 (metallothionein 2) was previously shown in a targeted study to have a dramatic diurnal rhythm [48]. Among genes ranked highly by LimoRhyde2 in the lung were Fkbp5 and Adamts4. The former encodes a negative-feedback regulator of glucocorticoid signaling [49], which plays an important role in synchronizing daily rhythms across tissues [50], while the latter is a clock-controlled gene in mouse cartilage [51] and human corneal endothelial cells [52]. Among genes ranked highly in the SCN were Nfkbib, which encodes an inhibitor of NF-κB signaling and whose expression is regulated in microglia by the clock gene Nr1d1 [53], and Id1, which may play a role in photic entrainment of the circadian system [54].

Despite its advantages, LimoRhyde2 still has limitations and opportunities for future improvements. First, we have only validated LimoRhyde2 for quantifying rhythmicity within a single condition, not for quantifying differences in rhythmicity between conditions. Second, LimoRhyde2 assumes that the rhythms have a user-specified period shared by all genomic features. Although it is possible to run LimoRhyde2 multiple times on the same dataset, varying the period each time, we recommend instead using a single period and allowing the periodic spline to capture non-monotonic (and potentially ultradian) rhythms. As with other methods,

**Table 2. Comparison of several methods for genome-scale rhythmicity analysis.**

|  | LimoRhyde2 | BooteJTK | CircaN | RAIN | dryR |
|---|---|---|---|---|---|
| Reference | this paper | [21] | [22] | [23] | [24] |
| Type of inference | effect size estimation | hypothesis testing | hypothesis testing | hypothesis testing | model selection |
| Type of model | parametric | non-parametric | parametric | non-parametric | parametric |
| Accounts for non-sinusoidal rhythms | ✓ | ✓ | - | ✓ | - |
| Handles unevenly spaced timepoints | ✓ | - | ✓ | ✓ | ✓ |
| Handles covariates | ✓ | - | - | - | - |

the ability to reliably detect and quantify ultradian rhythms will depend on the sampling interval and the signal-to-noise ratio. Third, LimoRhyde2 assumes that each feature's rhythm is fixed. Consequently, LimoRhyde2 does not model amplitude decay [55], although it can model time-dependent trends in mesor.

Just as prior methods cannot determine what level of adjusted p-value qualifies as "significant" (the conventional level of 0.05 being arbitrary), LimoRhyde2 cannot determine what magnitude of a given rhythm statistic is biologically meaningful. Such values likely vary from one gene to another anyway, so it remains possible that LimoRhyde2 could deemphasize biologically meaningful rhythms of low amplitude. When performing follow-up computational analyses, we recommend using the full distribution of a given statistic rather than drawing artificial cutoffs such as "rhythmic" and "arrhythmic". As a convenient reference for rhythm amplitudes, our results indicate that the core clock genes in wild-type mouse liver have a median peak-to-trough amplitude of ~3.3 ($log_2$CPM in RNA-seq). In addition, LimoRhyde2 cannot determine what range of credible interval is most biologically relevant. In the current analysis, we elected for simplicity and ranked genes only by the point estimates of amplitude. Future work can explore how credible intervals should inform follow-up analyses and experiments. However, no amount of statistical wizardry is likely to overcome low biological replicability [56].

By directly estimating biological rhythms and their uncertainty, LimoRhyde2 seeks to shift the focus of an analysis from detecting statistical significance to interpreting biological relevance. Although we have so far only validated LimoRhyde2 on bulk transcriptome data, recent work showed that the same state-of-the-art methods that underlie LimoRhyde2 are well-suited to analysis of single-cell RNA-seq data [43]. Thus, LimoRhyde2 may provide a basis for using various genomic techniques to improve our understanding of biological rhythms.

## Supporting information

**S1 Fig. LimoRhyde2 moderates amplitude based on standard errors of genes.** Scatterplots of difference between raw and posterior peak-to-trough amplitude vs $log_2$ mean standard error of the raw fit for genes in each tissue. *Points* represent genes.
(TIF)

**S2 Fig. LimoRhyde2's spline-based model is more flexible and tends to give higher amplitude than the cosinor model.** (**A**) Scatterplots of cosinor posterior peak-to-trough amplitude vs. spline posterior peak-to-trough amplitude for each tissue (indicated at *top*). *Points* represent genes. *Dashed lines* indicate y = x. (**B**) Time-courses of expression of genes labeled in (A) in the respective tissue (indicated at *right*). *Points* represent samples. *Curves* represent posterior fits for the two models.
(TIF)

**S3 Fig. LimoRhyde2 identifies generally strong rhythms of core clock genes.** Posterior peak-to-trough amplitudes and corresponding 90% credible intervals for core clock genes in each tissue. *Points* represent genes, *color* represents peak phase for each gene. *Dashed lines* indicate 0 amplitude.
(TIF)

**S4 Fig. Distributions of rhythmicity based on LimoRhyde2 posterior statistics.** Scatterplots of *(A)* peak-to-trough amplitude vs. mesor and (**B**) peak-to-trough amplitude vs. peak phase for genes in each tissue (indicated at *top*). *Points* represent genes. (**C**) Histograms of peak phase. All plots include only the top 25% of genes based on amplitude.
(TIF)

## Acknowledgments

We thank Layla Aref and Jeffrey Tatro for helpful comments on the manuscript.

## Author Contributions

**Conceptualization:** Dora Obodo, Elliot H. Outland, Jacob J. Hughey.

**Data curation:** Dora Obodo, Elliot H. Outland, Jacob J. Hughey.

**Formal analysis:** Dora Obodo, Elliot H. Outland, Jacob J. Hughey.

**Funding acquisition:** Jacob J. Hughey.

**Software:** Dora Obodo, Elliot H. Outland.

**Supervision:** Jacob J. Hughey.

**Validation:** Dora Obodo, Elliot H. Outland, Jacob J. Hughey.

**Visualization:** Dora Obodo, Elliot H. Outland, Jacob J. Hughey.

**Writing – original draft:** Dora Obodo, Elliot H. Outland, Jacob J. Hughey.

**Writing – review & editing:** Dora Obodo, Elliot H. Outland, Jacob J. Hughey.

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
