## [Decision Letter · Decision Letter 0]

21 Aug 2023

PONE-D-23-21973LimoRhyde2: genomic analysis of biological rhythms based on effect sizesPLOS ONE

Dear Dr. Hughey,

Thank you for submitting your manuscript to PLOS ONE. After careful consideration, we feel that it has merit but does not fully meet PLOS ONE’s publication criteria as it currently stands. Therefore, we invite you to submit a revised version of the manuscript that addresses the points raised during the review process.

We look forward to receiving your revised manuscript.

Kind regards,

Henrik Oster, Ph.D.

Academic Editor

PLOS ONE

Journal Requirements:

"HHS | NIH | National Institute of General Medical Sciences (NIGMS):Jacob J Hughey R35GM124685; HHS | NIH | National Institute of General Medical Sciences (NIGMS):Dora Obodo F31GM143909"

"We thank Layla Aref and Jeffrey Tatro for helpful comments on the manuscript. This work was supported by US NIH R35GM124685 to JJH and F31GM143909 to DO."

"HHS | NIH | National Institute of General Medical Sciences (NIGMS):Jacob J Hughey R35GM124685; HHS | NIH | National Institute of General Medical Sciences (NIGMS):Dora Obodo F31GM143909"

**Additional Editor Comments:**

Dear authors,

all three reviewers were quite positive about your paper. Few minor suggestions were raised, and I agree with reviewers 1 and 2 that performance in missing/unevenly spaced data point time series should be addressed. Further, a table comparing the specific strengths of some of the more established algorithms in comparison to LR2 would be appreciated.

Reviewers' comments:

Reviewer's Responses to Questions

**Comments to the Author**

1. Is the manuscript technically sound, and do the data support the conclusions?

Reviewer #1: Yes

Reviewer #2: Partly

Reviewer #3: Yes

2. Has the statistical analysis been performed appropriately and rigorously? 

Reviewer #1: Yes

Reviewer #2: Yes

Reviewer #3: Yes

3. Have the authors made all data underlying the findings in their manuscript fully available?

Reviewer #1: Yes

Reviewer #2: Yes

Reviewer #3: Yes

4. Is the manuscript presented in an intelligible fashion and written in standard English?

Reviewer #1: Yes

Reviewer #2: Yes

Reviewer #3: Yes

5. Review Comments to the Author

Reviewer #1: The study by Obodo et al provides a novel method to estimate rhythmicity. This new approach is highly welcomed, and the authors convincingly show the limitation of estimating rhythmicity by a fixed threshold (p-value) of rhythmicity with or without amplitude estimation. LimoRhyde2 bases on effect sizes estimation instead of p values for rhythmicity. The manuscript has also received criticism and suggestions from the Review Commons, which the authors have addressed. Therefore, my comments focus not on the methodological aspect but on a user’s perspective.

The manuscript would be improved if the authors provided a simple table in which it is mentioned the pro and cons of LimoRhyde2 against established methods just as JTK, BooteJTK, RAIN, DryR, CircaN, etc.

There is an increased interest in the field of detecting ultradian rhythms. There are few tools properly established for this detection. The manuscript would improve if the authors provided more detail on how LimoRhyde2 can be used (if at all) to detect ultradian rhythms.

The authors heavily criticize using p- or q-values to establish rhythmicity. I do not intend to over-discuss this issue, but the authors focus on amplitude instead. However, several potential candidates can be excluded by selecting only high-amplitude rhythms. This could lead to the loss of several potentially rewarding targets for follow-up analysis. Mild or low amplitude rhythmic genes can still show robust rhythms at the protein level and vice-versa. The should discuss this in brief.

In addition to amplitude, phase is an essential parameter for circadian studies. Phase assessment is not discussed in the manuscript. The manuscript would improve if the authors provided some evidence for phase assessment compared to established methods.

Recently Brooks et al (10.1177/07487304231179600) showed concerning findings on the lack of reproducibility of circadian/diurnal transcriptome across several studies. In particular, the authors only focused on phase as amplitude was highly variable across the studies. Considering that LimoRhyde2 method is based on amplitude estimation, how trustworthy would this method be to assess rhythmicity across similar studies? The authors should discuss this in the manuscript.

Reviewer #2: This revised manuscript addressed the issues raised by the previous 3 reviewers and described an interesting new method to characterize rhythmicity of gene expression. The limitations of the method are well described in the discussion. Nevertheless, one limitation still needs to be addressed. As stated by the authors in the introduction, many methods rely on timepoints being equally spaced and cannot cope with missing data. However, authors did not provide evidence that LimoRhyde2 can perform well in such conditions and even replaced missing values by “non-zero counts” from other time points in the chosen mouse datasets. Therefore, it could be great if the authors could expand their validation (for example using simulated data) by testing alternative sampling frequencies (2 hour, 4 hour, 6 hour), sampling durations (1 cycle vs 2 cycles, vs partial cycle), replicate numbers, and in datasets with missing timepoints/values.

Reviewer #3: I am happy with authors' response to my comment in the first round (via Review Commons). I do not have any additional comments for the authors.

Thank you for taking the time to address my previous concerns.

6. PLOS authors have the option to publish the peer review history of their article (what does this mean?). If published, this will include your full peer review and any attached files.

Reviewer #1: No

Reviewer #2: No

Reviewer #3: No

---

## [Author Response · Author response to Decision Letter 0]

8 Sep 2023

See attached responses to reviewers.

---

## [Editor Report · Decision Letter 1]

12 Sep 2023

LimoRhyde2: genomic analysis of biological rhythms based on effect sizes

PONE-D-23-21973R1

Dear Dr. Hughey,

We’re pleased to inform you that your manuscript has been judged scientifically suitable for publication and will be formally accepted for publication once it meets all outstanding technical requirements.

Kind regards,

Henrik Oster, Ph.D.

Academic Editor

PLOS ONE

Additional Editor Comments (optional):

Congrats on this nice paper.
---

## [Editor Report · Acceptance letter]

18 Sep 2023

PONE-D-23-21973R1 

LimoRhyde2: genomic analysis of biological rhythms based on effect sizes 

Dear Dr. Hughey:

I'm pleased to inform you that your manuscript has been deemed suitable for publication in PLOS ONE. Congratulations! Your manuscript is now with our production department. 

Kind regards, 

on behalf of

Prof. Henrik Oster 

Academic Editor

PLOS ONE